# Agility Testing in Youth Football (Soccer)Players; Evaluating Reliability, Validity, and Correlates of Newly Developed Testing Protocols

**DOI:** 10.3390/ijerph17010294

**Published:** 2020-01-01

**Authors:** Ante Krolo, Barbara Gilic, Nikola Foretic, Haris Pojskic, Raouf Hammami, Miodrag Spasic, Ognjen Uljevic, Sime Versic, Damir Sekulic

**Affiliations:** 1Faculty of Kinesiology, University of Split, 21000 Split, Croatia; ante.krolo@gmail.com (A.K.); barbara.gilic@outlook.com (B.G.); nikolaforetic@gmail.com (N.F.); mspasic@kifst.hr (M.S.); ognjen.uljevic@gmail.com (O.U.);simeversic@gmail.com (S.V.); damirsekulich@gmail.com (D.S.); 2Faculty of Kinesiology, University of Zagreb, 10000 Zagreb, Croatia; 3Department of Sports Science, Linnaeus University, 39182 Kalmar, Sweden; 4Research Unit, Education, Motricity, Sport and Health, UR15JS01, High Institute of Sport and PhysicalEducation of Sfax, University of Sfax, Sfax 3000, Tunisia; raouf.cnmss@gmail.com

**Keywords:** team sports, pre-planned agility, non-planned agility, determinants, soccer

## Abstract

Reactive agility (RAG) and change of direction speed (CODS) are important determinants of success in football (soccer), but there is an evident lack of information on reliable and valid football-specific testing procedures which will be applicable in defining sport-specific RAG and CODS in youth players. This study evaluated reliability and construct validity of newly developed tests of football-specific RAG (FS_RAG) and CODS (FS_CODS), which involved the ball kicking football technique. Additionally, factors associated with FS_RAG and FS_CODS were evaluated. The participants were youth football players (n = 59; age: 13.40 ± 1.25 years) divided according to their age into U13 (11–12 years of age; n = 29), and U15 (13–14 years of age; n = 30) categories. Additionally, performance levels (starters [first-team] vs. non-starters [substitutes]) were observed in each age category. The dependent variables were newly developed FS_RAG and FS_CODS tests. The independent variables were sprinting capacities over 10 and 20 meters (S10M, S20M), countermovement jump (CMJ), the reactive strength index (RSI), and a generic CODS test of 20 yards (20Y). The newly developed FS_CODS and FS_RAG were observed as dependent variables. Results showed appropriate intra-testing and inter-testing reliability of the FS_RAG and FS_CODS, with somewhat better reliability of the FS_CODS (ICC=0.82 and 0.79, respectively). Additionally, better reliability was evidenced in U15 than in U13 (ICC: 0.82–0.85, and 0.78-0.80 for U15 and U13, respectively). Independent samples t-test indicated significant differences between U13 and U15 in S10 (*t*-test: 3.57, *p* < 0.001), S20M (*t*-test: 3.13, *p* < 0.001), 20Y (*t*-test: 4.89, *p* < 0.001), FS_RAG (*t*-test: 3.96, *p* < 0.001), and FS_CODS (*t*-test: 6.42, *p* < 0.001), with better performance in U15. Starters outperformed non-starters in most capacities among U13, but only in FS_RAG among U15 (*t*-test: 1.56, *p* < 0.05). Multiple regression calculations indicated nonsignificant association between independent and dependent variables in U13 (FS_CODS: 19%, FS_RAG: 21% of the explained variance, both *p* > 0.05), but independent variables explained significant proportion of both dependent variables in U15 (FS_CODS: 35%, FS_RAG: 33% explained variance, both *p* < 0.05). The study confirmed the applicability of newly developed tests in distinguishing studied age categories of players. Results indicate that superiority in all studied fitness capacities is translated into performance level in U13. Meanwhile, FS_RAG seems to be important determinant of quality in U15.

## 1. Introduction

Agility is defined as a rapid change of speed and direction of movement as a reaction to an external stimulus [1]. This quality is recognized as an important health-related fitness component[2], and an essential quality in some professional activities (i.e., military, police) [3,4]. However, agility is mostly known as crucial conditioning capacity in competitive team sports, including football [5,6,7]. 

Football (soccer) is an intermittent team sport characterized by frequent transitions in activity patterns, from high-intensity movements such as sprinting, jumping, shooting, accelerating, and decelerating to low-to-moderate intensity activities such as jogging, walking and even standing still [8,9]. Demands of the football game are increasing rapidly; thus players have to possess a high level of physical capacities, technical and tactical skills [10]. However, ability to rapidly change speed (i.e., acceleration) and sprint is the most frequent action in goal situations in professional football [11] . The observed accelerations are mainly performed in response to external stimuli (i.e., movement of the ball, opponent, teammate) and usually preceded by change of movement direction. This essential football ability, described in such way, represents, in fact, agility.

There area largenumber of tests and testing protocols used to measure agility performance in team sports, including football [12,13]. However, most of these tests actually test only ability to quickly change the direction (i.e., change of direction speed—CODS), without including the response to external unpredictable stimuli, which is essential facet of agility. Therefore, a clear distinction between CODS (i.e., pre-planned agility or non-reactive agility), and reactive agility—RAG (i.e., non-planned agility) is required, especially in detecting cognitive (i.e., perceptual, decision making) and physical (i.e., conditioning capacities, body indices, technique etc.) related determinants of agility[14]. Indeed, while various conditioning capacities (i.e., sprinting speed, jumping performance) were more strongly associated with CODS than with RAG, perceptual and cognitive capacities were identified as being significant predictors of RAG in team sport athletes [14,15]. Meanwhile, there is an evident lack of studies which examined factors associated with CODS and RAG in football. These information may be of high importance in training and conditioning since it will allow specific and targeted development of important qualities which will consequently improve specific RA or CODS quality [16].

Because of the known importance of agility for long-term sport development, studies frequently elaborated the reliability and validity of the tests aimed at evaluation of agility in young athletes[17,18]. However, mostof the investigations done so far evaluated CODS-, and not RAG-performances in youth athletes involved in different sports, including football. For example, Basque authors examined the reliability and validity of the modified Barrow CODS test and reported high test-retest reliability (ICC: 0.94) for 11-to-14 years old football players[19]. Similarly, Tunisian authors reported appropriate reliability of the zig-zag CODS test in 14.2 ± 0.6 years-old players [20]. Finally, the T-drill CODS test had high reliability in 10–12 years old football players [21]. However, there is an evident lack of information about reliability of the RAG tests in football. 

With regard to agility in football, it is also important to note that performance in this sport often involve dribbling, passing, and kicking movements, indicating the importance of ball-handling even in CODS and RAG[11]. As a result of this and importance to mimic the situations which appear in sport (i.e., face validity), there is an increase in popularity of football-specific tests (i.e., tests designed to simulate real-game situations in football). Not surprisingly, researchers have developed even football-specific CODS and RAG tests including a certain type of ball handling[22,23].Benvenuti et al. proposed a valid and reliable testing protocols in defining sport specific RAG in female football and futsal players [22].More recentlyPojskic et al. investigated football-specific RAG and CODS tests, where players performed agility tests with adding the football-specific movement (i.e., kicking the ball). In brief, tests showed high reliability and power to discriminate between playing levels (i.e., U17 vs. U19 players) [23].However, to the best of our knowledge no study proposed test procedure aimed at evaluation of football-specific agility in younger players (i.e., younger than 15 years). 

The aims of this study were: (i) to evaluate the reliability and construct validity of the football specific test of RAG (FS_RAG) and football specific test of CODS (FS_CODS), and (ii) to identify associations between specific conditioning capacities with FS_RAG and FS_CODS in youth football players. The leading hypotheses of the research were: (i) the newly developed tests will have proper reliability and validity in distinguishing age groups and performance levels of studied players, and (ii) the fitness capacities will be significantly correlated to FS_RAG and FS_CODS in studied players. 

## 2. Materials and Methods 

### 2.1. Study Design 

This study combined test–retest (repeated measurement) and cross-sectional approaches, and consisted of several phases. Initially, the design of the newly developed tests of football-specific RAG and CODS (FS_RAG and FS_CODS), which are applicable in youth players, was determined through consultations with three experts in football, who were informed about the main idea, previously applied test designs in football and other team sports, technical requirements, boundaries, etc. Finally, the testing design and scenarios proposed previously for older football players (i.e., U17 and U19) were simplified and applied herein (see “Variables” for details) [23]. 

The repeated measurement(separated by 7–9 days), was applied in order to define the reliability of the developed RAG and CODS tests, which was additionally broadened in further cross-sectional measurements of independent variables, consisting of basic anthropometrics, sprinting, generic-CODS, and jumping capacities. Specifically, groups of participants (age groups and performance levels) were compared on all applied variables in order to identify the construct validity of the tests. Finally, independent variables were correlated to dependent variables in order to identify predictors of football-specific CODS and RAG.

### 2.2. Participants

Participants in this study were young football players (all males, n = 59, age: 13.40 ± 1.25 years, body height: 162 ± 8.14 cm, mass: 54.1 ± 9.15 kg, body mass index: 20.57 ± 2.11 kg/m^2^) from two football academieslocated in Split-Dalmatia county in southern Croatia. Football academies included in this study were selected on the basis of following criteria: (i) having football teams in all age categories starting from U11 to seniors (+18 years), (ii) having at disposal courts with artificial-plastic turf (see later for details about testing), and (iii) licensed (professional) coaches working with U13 and U15 teams (see later for details).The total sample of players was divided into two subsamples according to their age: (i) players who were 11–12 years of age (U13; n = 29) and players who were 14–15 years of age at the moment of testing (U15; n = 30). For the purpose of this study, players in each age group (U13 and U15) were additionally grouped, according to their performance level, into starters (14 and 15 players in U13 and U15, respectively) and non-starters. The grouping into starters and non-starters was done by coaches, and investigators had no influence on coaches’ decision. Specifically, starters were players who regularly “start the game” (i.e., better players as defined by their coaches), and non-starters were actually their “substitutes”. In this study, we observed only field players, meaning that goalkeepers were not observed. All participants had at least 3 years of experience in systematic football training, and at the moment of testing participated in 3-4 training sessions weekly, plus one game. Most of their trainings (more than 80%) was oriented toward specific technical and tactical skills, but two of 10 training sessions are oriented toward development of physical and conditioning capacities. Players from both football academies were equally represented in each age group. For all participants, pubertal timing was estimated according to the biological age of maturity (maturity offset), as described by Moore et al. (Maturity offset = −7.999994 + (0.0036124 × age (yrs.) × height (cm)[24]. 

While all participants (n = 59) were included in cross-sectional measurement (see previous text for details), in the repeated measurement (test–retest procedure) we observed 14 players to define the inter-testing reliability. Test and retest were separated by 7–9 days. Prior to the study, participants and their parents/responsible adults were informed about the purpose, design, benefits, and risks of the study, and parents provided informed consent for the study participation of their children. The study was approved by ethical board of the University of Split, Faculty of Kinesiology (Ethical Board Approval No: 2181-205-02-05-14-001). The testing was done during March 2019, and all to avoid diurnal variations, all tests were performed in the morning hours between 10:00 and 12:00.

### 2.3. Variables

Independent variables consisted of anthropometrics (body mass, height), one generic CODS test, two tests of sprinting performance, and two tests of jumping performances. Dependent variables were two newly designed tests of football-specific CODS (FS_CODS) and football-specific RAG (FS_RAG). Participants were familiarized with dependent variables a day before testing, and performed several testing trials at submaximal intensity and 1–2 at maximum. The familiarization trials were not included in further analyses. Since participants were already tested on independent variables, the familiarization was not necessary. 

Anthropometrics were measured by an experienced technician using standardized equipment. Body mass (BM) was measured in 0.1 kg, and body height in cm (extrapolated at 0.5 cm). The countermovement jump test (CMJ) was conducted with an athlete starting from an upright position with hands on the hips, performing fast downward movement to approximately 90° of knee flexion followed by a maximum-force upward vertical motion. The height and the contact time of the jump were measured by the Optojump system (Microgate, Bolzano, Italy)[14]. Test was performed over three trials, with 1 min rests between trials, and the best performance was considered the final achievement for each participant.

The reactive strength index (RSI) was calculated from the height jumped, and from the contact time with the ground while performing the double-leg drop (depth) jump. For the start of the drop jump, athletes were standing on a 40 cm high box and were instructed to step off forward between two photoelectric beams (Optojump system) and to jump upwards maximally while minimizing the contact time[14]. The test was performed over three trials, with 2 min rests in between, and the best performance indicator (the highest numerical value) was used as the final result for each participant. 

For the purpose of this study, we included a twenty yards test (20Y) in the test of generic CODS capacity[25]. The test was conducted on the 10-yard field, with one cone placed in the middle, one cone placed 5 yards on the left side, and one cone placed 5 yards on the right side. The athlete was standing in the lateral stance 50 cm from the middle cone, where the timing gate (Powertimer, Newtest, Oulu, Finland) was placed. The test started with an athlete rotating for 90° and triggering the timing gate by running 5 yards to the left, turning and sprinting 10 yards to the right, and turning and sprinting back to the middle line when the time recording finished. Athletes performed three trials with 2–3 minutes of rest between the trials, and the best attempt was included in the analyses. 

Sprinting was assessed by a sprint test of 10 m (S10M) and 20 m (S20M). One photoelectronic timing gate (Powertimer) was placed at the start line, a second gate was placed at 10 m from the marked starting line, and a third at 20 m, with reflectors placed at a 1 m height. From the standing start, the athletes had their preferred foot placed forward on the marked line 1 m before the start line (to avoid switching on the timing gate accidentally by arm movements) and were instructed to run at maximum speed along the 25 m field. The time was recorded at 10 m (S10M) and 20 m (S20M). Athletes performed three testing trials, with 2–3 minutes of rest between the trials, and the best achievements were used for the analyses.

The FS_CODS and FS_RAG were measured using a hardware device system based on an ATMEL micro-controller (model AT89C51RE2; ATMEL Corp, San Jose, CA, United States), which has been previously used in similar studies in handball, basketball, futsal, and football [14,23,26,27]. A photoelectric infrared (IR) sensor (E18-D80NK) was used as the time triggering input, and LEDs placed in the 30-cm-high cones were used as controlled outputs. Both tests had similar patterns (see Figure 1) with specified distances, and after reliability analyses (see later for details), the best achievement was used as final result for each participant. 

The FS_RAG test started with the player running at maximal intensity through gate where the IR signal was placed (Figure 1). At that moment, the timing began, one of the cones (either “A” or “B”) was lit, and the player had to run at maximum speed in the direction of that cone, kick the ball with the inside of the foot on the goal, and turn back through the starting gate as quickly as possible. By passing the starting gate on the way back, the timing stopped, and the running speed was recorded. The attempt was repeated if the ball did not pass throughout goal line. For FS_RAG, players did not know which cone would be lit after passing the starting gate, so they had to react to an unpredictable stimulus and then perform the necessary action with the most appropriate pattern. All players performed five trials with no advanced knowledge of the testing scenario. 

For the FS_CODS test, players were tested on the same testing field (Figure 1), but with advanced knowledge about the testing scenario (i.e., they knew which cone would be lit and therefore did not have to react on unpredictable visual stimuli as for the FS_RAG). All players were tested on the same pattern and performed two trials, first running in the direction of the cone placed on the left side (Cone “A”) and secondly to the right side (Cone “B”). In other words, when tested on FS_CODS, they were able to pre-plan the running direction. 

### 2.4. Statistical Analysis

All data were log-transformed to reduce the non-uniformity of error, and all statistical analyses were done on transformed data. Distributions of the variables was checked by Kolmogorv Smirnov’s test (Appendix A), and results are presented in raw data (non-transformed) data, with means and standard deviations reported. 

The inter-testing reliability of the FS_CODS and FS_RAG was analyzed by the intraclass correlation coefficient (ICC; model 3,1) as a measure or relative reliability, and by the coefficient of variation (CV) as a measure of absolute reliability [28,29,30]. The standard error of measurement (SEM) was also calculated (SEM = SD −√(1 − ICC)).

Additionally, all variables were checked for intra-testing reliability, with testing items treated as repeated measures for each variable. First, a repeated measures ANOVA with a corresponding Tukey post hoc test was calculated for each variable to assess systematic error (e.g., fatigue and learning) among trials, and the intra-testing ICCs and CVs were then calculated for all variables [31,32]. 

The construct validity of the tests was assessed by comparing (i) age groups (i.e., U13 vs. U15) and (ii) performance levels (starters vs. nonstarters) in each age group. For such purposes, a 2-sided t-test for independent samples was calculated. Additionally, Cohen’s d effect sizes (ES) for differences in studied variables between corresponding groups (e.g., U13 vs. U15; starters vs. non-starters) were calculated, and they were interpreted using the following qualitative descriptors: <0.2 = trivial, 0.21–0.49 = small, 0.50–0.79 = medium, >0.79 = strong [33]. 

To establish the univariate associations between variables, Pearson’s product moment correlation (r) was calculated. The standard multiple regressions were calculated to identify the multivariate relationships between independent variables (predictors) and dependent variables (criteria; FS_RAG and FS_CODS). For the purpose of the interpretation of explained variance, the coefficient of multiple correlation, and the coefficient of determination (percentage of explained variance) were calculated. In order to identify partial contribution of each predictor to overall multiple regression calculation, the standardized- and non-standardized-regression-coefficients were reported. Before multiple regressions’ calculation, the predictors were checked for multicollinearity by calculating the variance inflation factor (VIF), and those variables with a VIF larger than 10 were not included as predictors in the specific multiple regression calculation [34,35,36,37]. 

The *p*-level of 95% was applied, and Statistica v.13.0 (Dell Inc., Palo Alto, CA, USA) was used for all statistical analyses. 

## 3. Results

The inter-testing reliability of FS_CODS and FS_RAG was appropriate (ICC: 0.79 and 0.82, CV: 5% and 4%, for FS_RAG and FS_CODS, respectively), with better reliability of FS_CODS than of FS_RAG (Table 1). This was additionally confirmed when intra-testing reliability was calculated separately for U13 and U15 Additionally, ICC and CV showed better reliability of FS_RAG and FS_CODS for U15 than for U13 players (Table 2). 

The U15 players achieved significantly better results than U13 inS10M (t-test: 3.57, *p* < 0.001; large ES differences), S20M(t-test: 3.13, *p* < 0.001; large ES differences), 20Y (t-test: 4.89, *p* < 0.001; large ES differences), FS_RAG (t-test: 3.96, *p* < 0.001; large ES differences), and FS_CODS (t-test: 6.42, *p* < 0.001; large ES differences) (Table 2; Figure 2). 

The differences between performance levels (starters vs. nonstarters) in the tested variables for each age category are presented in Table 3 and Figure 3. In the younger age group (U13), starters were more advanced in biological age (maturity offset), and outperformed non-starters in all capacities, but RSI (t-test:0.89, *p*> 0.05), with a large ES for maturity offset, S10M, S20M, 20Y, FS_RAG, and moderate ES for CMJ and FS_CODS. 

The only variable that significantly differentiated performance groups in the older group (U15) was FS_RAG (t-test: 1.56, *p* < 0.05; moderate ES differences), with a better performance in starters than in nonstarters. Pearson’s product moment correlation coefficients did not reach statistical significance when correlations were calculated between independent and dependent variables in U13.

Of the 12 calculated correlations, eight were statistically significant in U15. Specifically, performance in FS_RAG and FS_CODS was associated with performance in S20M (r = 0.48 and 0.38), achievement in 20Y (r = 0.63 and 0.50), and CMJ results (r = 0.57 and 0.59 for FS_RAG and FS_CODS, respectively) (Table 4). 

Multiple regressions for FS_RAG and FS_CODS criteria were not significant when calculated for U13 (19% and 21% of the explained variance for FS_CODS and FS_RAG, respectively, both *p*> 0.05) (Table 5). When multiple regression was calculated between predictors and criteria for U15, the better FS_RAG in U15 players who had better generic CODS (20Y), and superior vertical jumping capacities (35% of explained variance). Also, similar structure of predictors contributed significantly to FS_RAG in U15 players, with 33% of the explained variance (Table 6).

## 4. Discussion

There are several important findings of this study. First, the reliability of the newly developed tests of football-specific CODS and RAG is appropriate, with better reliability of FS_CODS in both age groups. Next, the discriminative validity (construct validity) of the newly developed tests differs across the studied age groups. Additionally, conditioning capacities are significantly correlated to FS_CODS and FS_RAG in U15 but not in U13. Collectively, we may accept our first study hypothesis (that FS_CODS and FS_RAG have appropriate reliability and validity). Because of the week correlations between studied conditioning capacities with FS_CODS and FS_RAG, the second study hypothesis is denied.

### 4.1. Reliability

Inter-testing and intra-testing reliability of the FS_RAG and FS_CODS is comparable to tests of similar capacities in other sports, and football-specific RAG and CODS in older male and female players [22,23,27]. In brief, when studied professional basketball players on basketball-specific CODS and RAG, Pehar et al. reported values of 0.80 for intra-testing reliability [14]. Further, the reliability of FS_RAG and FS_CODS in our study was similar to reliability of tests aimed to determine same type of capacities in older male and female football players [22,23]. 

Specifically, Benvenuti et al. observed Italian female players and reported ICC of 0.80 for specific football RAG test, while Pojskic et al. recently presented results of the study with U17 and U19 male players from Sweden and reported 0.70–0.90 for inter- and intra-testing reliability [22,23]. It must be mentioned that in here presented investigation, authors intentionally simplified the testing sequence proposed in cited study of Pojskic et al. (which included four directional possibilities), and included only two directional possibilities (see Figure 1) [23]. Such difference in testing design reduced the possibility of making mistake during testing in here tested younger players, increased the correlation between testing trials (both, intra- and inter-testing), which altogether positively influenced the reliability despite the participants’ younger age. Additionally, we must not ignore the fact that several familiarization trials which were applied testing (see Methods for details), almost certainly assured identification of the most proper movement pattern for each tested player, and assured stability of testing results.

The fact that the reliability of the FS_CODS is better than the reliability of the FS_RAG is consistent with previous reports where authors simultaneously observed different types of CODS and RAG tests [26,38]. In brief, in the study done with professional basketball players, the reliability of the basketball-specific CODS was higher than reliability of RAG test performer on the same course and with corresponding movement template [39]. Next, similar results (i.e., better reliability of CODS than of the RAG test) are presented in handball study, both in male and female players [26], male futsal [27] and male football players [23]. The findings are regularly explained by higher complexity of RAG than of CODS tests. In brief, while both CODS and RAG performances depend on similar conditioning capacities (i.e., speed, power, balance), the perceptual and cognitive capacities are important determinants of RAG only [15]. While each determinant (i.e., factor of influence) of the RAG and CODS presents the theoretical source of measurement error, the relatively lower reliability of RAG tests is understandable. Supportively, even studies which examined exclusively CODS tests reported lower reliability of tests which consisted of more changes of direction, when compared to tests consisting of fewer changes of direction, and/or less complex scenarios [25]. 

### 4.2. Validity

The newly developed FS_CODS and FS_RAG are found to be applicable in defining the differences between U13 and U15. Therefore, we may accentuate the construct validity of the tests in defining performance levels in youth age football players. However, age groups significantly differed in sprinting, and generic CODS as well (for details, see Table 2). Therefore, when it comes to differentiation between the age groups observed here (U13 and U15), the construct validity of the newly developed FS_RAG and FS_CODS is similar to the validity of sprinting and generic CODS tests. 

CODS and RAG of youth age football players are most likely not fully differentiated from other conditioning capacities. Indeed, it is well documented that clear differentiation between various motor capacities occurs later in puberty [40]. Therefore, it is probable that the level of general conditioning status is even translated to a status of CODS and RAG in studied players. The previous discussion is supported even in established differences between starters and nonstarters in U13, where starters achieved better results in most of the studied fitness capacities and were advanced in biological maturity. However, only FS_RAG significantly distinguished starters from non-starters in U15. In explaining such a finding, we must shortly overview the problem of biological maturity status and its influence on conditioning capacities in youth [41,42]. 

In brief, it is well documented that maturity status in puberty is one of the most important determinants of physical, but also psycho-social development [43]. This is particularly evident in boys, since early maturing boys tend to experience a more intense adolescent growth spurt, resulting in greater pubertal gains in body mass, height, and lean body mass [41,42]. For these reasons, early maturing boys may even expect (potential) athletic advantages. Specifically, the intensive development of body size may contribute to improved power, strength, and speed. This is particularly possible in the early pubertal period (i.e., between the ages of 11 and 14 years), since in this period of life the differences in bodily size and function are the most evident [44]. Therefore, for those sports where speed, power, and strength are important determinants of competitive achievement, maturity status plays an important role, particularly between 11 and 14 years of age. Therefore, it is not surprising that starters in the younger age group (12–13 years of age) were biologically advanced in maturation status (please see differences in maturity offset), and outperformed the non-starters in all studied capacities, including FS_RAG and FS_CODS. The previous discussion may be additional supported by the fact that correlation between FS_CODS and FS_RAG is stronger in U15 than in U13 (Pearson’s r =0.42 and 0.58, respectively), which is altogether at least partially related to maturation process. 

While maturation level is an important determinant of overall conditioning status in the younger age group (U13) and consequently is directly translated to their performance level, our results highlight that skill level becomes more important in the older age group (14–15 years of age). The RAG performance is generally considered as being more influenced by the level of expertise in specific motor skills than the corresponding CODS [14]. Specifically, while achievement in the majority of CODS tests was related to sprinting and power capacities, the achievement in the corresponding RAG tests was regularly connected to perceptual and cognitive capacities, as well as the skill level of the tested athletes [15,25]. The fact that only FS_RAG differentiated performance-level groups in the older age group is a logical consequence of the fact that, in U15, participants do not differ in maturity status as much as their younger peers in U13. 

### 4.3. Correlates of Reactive Agility and Change of Direction Speed 

The FS_RAG and FS_CODS shared 16% of the common variance in younger players and 35% of the common variance in older players, and we may highlight the relative independence of these qualities in youth football players. This is supportive of previous studies where authors have confirmed similar correlations between corresponding CODS and RAG tests in college-level athletes (up to 36% of the common variance) [38,45], handball players (20% of the common variance) [26], and basketball players (up to 25% of the common variance) [39]. Although CODS and RAG tests have some similar background, the vast majority of its variance depends on independent qualities, even when both performances are tested using the same movement template [45]. This is additionally supported by the fact that, in this study, numerical values of correlation coefficients between FS_RAG and FS_CODS are very similar to the correlation that was evidenced between the generic CODS test (20Y) and FS_RAG. 

The implicit goal of the CODS and RAG tasks is to redirect total body momentum in a new direction as quickly as possible, so it is expected that, despite the purported importance of decision making, the physical actions constitute a great proportion of time necessary to complete CODS and RAG tasks [16]. However, running speed and jumping performances are generally considered stronger determinants of CODS than of the corresponding RAG capacities, and this is mostly explained by the fact that RAG performances regularly include perceptual and cognitive processing, so the influence of “fitness” variables is not so pronounced [46]. However, in our study, there is no evident difference in the percentage of explained variance when predictors were correlated to FS_CODS (35% of the common variance) and to FS_RAG (33% of the common variance). This is almost certainly the consequence of testing design and the sport-specific movement template included in the testing sequences of FS_RAG and FS_CODS. Supportively, when Pehar et al. examined correlates of basketball-specific CODS and RAG, the studied predictors showed similar percentages of variance in the RAG and CODS tests (35% and 40%, respectively) [14]. On the other hand, linear sprint and jumping performances were much better correlated with CODS than with RAG when investigators observed generic tests (i.e., not sport-specific tests) [15] and/or tested athletes involved in different sports (i.e., college athletes) [46]. 

To the best of our knowledge, only one study so far has examined the factors associated with RAG in football players. Specifically, Lloyd et al. observed young football players and examined the relationships between functional movement screen (FMS) scores, maturation, and various physical performances (including reactive agility) in young football players [47]. Since in this study authors did not report correlates of reactive agility other than the variables derived from FMS, their results are unfortunately incomparable to those presented here. Meanwhile, f et al. reported a stronger correlation between RSI and reactive agility (30% of the common variance) in senior basketball players than found herein, but this may be attributed to differences in the age of participants (12–15 years and 21 ± 3 years of age for football players and basketball players) and the consequent stronger influence of physical capacities on reactive agility performance in older age. 

Indeed, the observed predictors explained a larger percentage of variance of criteria (FS_RAG and FS_CODS) in the older group than in the younger group, and this is consistent with the recent report of Hammami et al., where authors investigated predictors of CODS performances in youth handball players [48]. Most likely, the younger group lacked the specific skills necessary to perform CODS and RAG maneuvers effectively, and therefore were not able to incorporate their sprinting and jumping capacities in FS_RAG and FS_CODS. As a direct consequence of longer involvement in football and systematic training, the older group has a higher level of skills, which directly allows them to perform FS_RAG and FS_CODS effectively while incorporating the necessary conditioning capacities. In other words, the older players were able to perform FS_RAG and FS_CODS at maximum intensity and to use their sprinting and jumping potential to a greater extent than their younger peers. 

The previous discussion can be translated into strength and conditioning processes in youth football. In short, the stronger association between the studied conditioning qualities and FS_RAG and FS_CODS in older boys indicates that the training process in younger players should be mostly focused on the development of specific motor skills, techniques, and corresponding neural parameters. While this is important for the long-term development of young players, it will probably also facilitate an improvement in technical competency and assist in coping with anthropometric changes that occur in this period of life [49]. For this purpose, we may suggest the usage of different forms of speed–agility–quickness (SAQ) training, as well as small-sided games. On the other hand, our results suggest that, in the older group, the focus on strength and conditioning should instead be placed on the development of sprinting and jumping capacities, which have been found as important determinants of both FS_RAG and FS_CODS in 14–15-year-old football players. 

### 4.4. Limitations and Strengths

The main limitation of this study is its cross-sectional design. Therefore, in order to clearly identify the cause–effect relationships between studied variables, intervention studies are needed. Further, participants were selected from two football teams; therefore, we cannot ignore the possibility that differential training program influenced our findings to some extent. However, in both groups (older and younger) we included a similar number of participants from two teams, which probably reduced the possibility of a previously noted bias. Also, all tests were performed on artificial turf (artificial grass), and since players perform on natural turf, there is a certain possibility that they did not perform at maximal level due to non-familiarity. However, usage of the artificial turf was necessary to assure standardized (i.e., similar) testing conditions for all participants. Finally, players were not observed according to playing position, which will almost certainly contribute to better applicability of the results. However, U13 players were not strictly defined with regard to their playing-position in football game, which consequently did not allow us to apply such grouping into experimental design. 

This is likely one of the first studies to examine the reliability and validity of football-specific tests of RAG and CODS, specifically in youth age, and likely the first in which two age categories were separately observed. A very narrow age span in each of the studied groups is also a substantial strength of this research. Finally, tests were based on previous research in the field and on experiences in the development of similar tests for other sports and for other age groups in football. Therefore, we believe that, although this investigation is not the final word on the problem, it will improve the knowledge in the field and that further research will be initiated. 

## 5. Conclusions

The football-specific tests applied in this study showed appropriate reliability and therefore may be used as appropriate and consistent testing procedures in evaluation of RAG and CODS in youth football players. However, it must be stressed that the study consisted of familiarization and testing trials, so we may accentuate the necessity of familiarization with the testing sequence before testing both capacities. 

When it comes to the validity of the tests in distinguishing the age groups studied herein (U13 vs. U15), the validity of the FS_RAG and FS_CODS is comparable to the validity of the tests evaluating sprinting, jumping, and generic CODS. However, the high validity and consequent applicability of the FS_RAG is evidenced in distinguishing performance levels in U15. Therefore, it can be concluded that, while physical and maturity-related variables are more important determinants of success in younger age, with the advancement of the maturity process, the importance of skill-related variables increases. 

From the practical aspect it is important to note that that our results point that improvement in sprint and jumping could be theoretically beneficial in improvement of sport-specific RAG and CODS in U15 players. Meanwhile, the RAG and CODS in U13 players are more influenced by various neurological factors, including technique and quality of the execution of the specific movement templates that occur in the applied testing sequence. As a result, training programs aimed at the improvement of RAG and CODS in early pubescent football players should be more oriented toward achieving an accurate and effective movement technique, and not toward the development of conditioning capacities, which (theoretically) contribute to RAG and CODS due to physiological background (i.e., the necessity of the fast development of force). 

## Figures and Tables

**Figure 1 ijerph-17-00294-f001:**
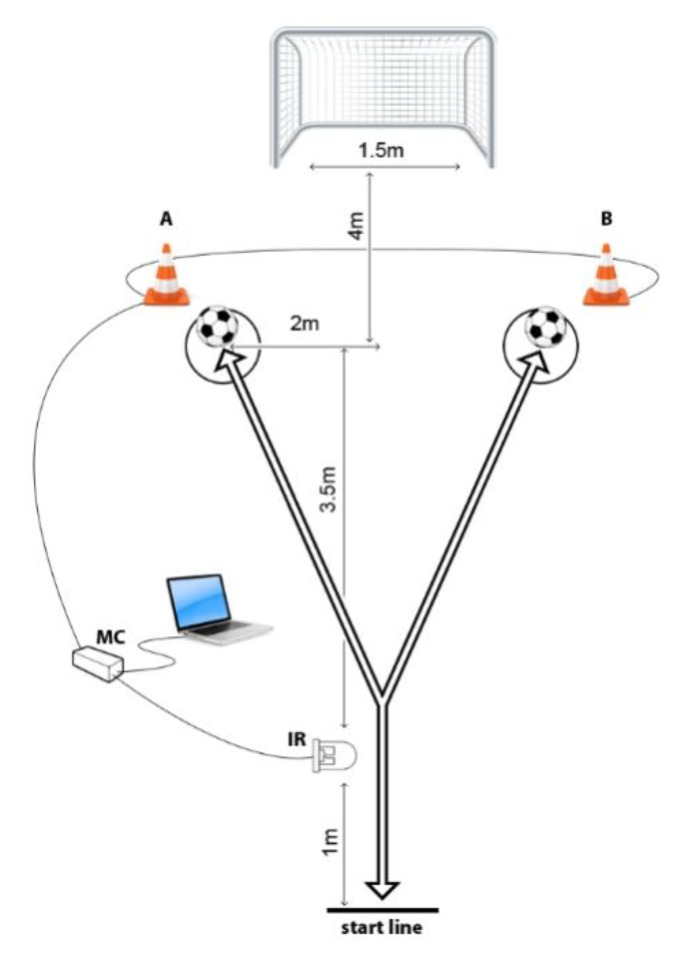
Football/specific reactive agility and change of direction testing polygon.

**Figure 2 ijerph-17-00294-f002:**
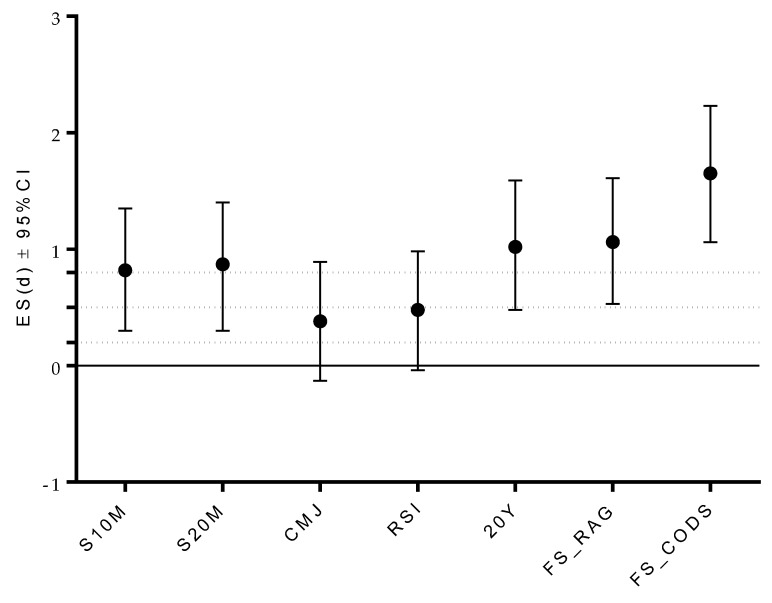
Effect size (ES) differences in studied variables between U13 and U15 age-category. LEGEND: S10—sprint 10 meters, S20M—sprint 20 meters, CMJ—countermovement jump, RSI—reactive strength index, 20Y—change of direction speed 20 yards test, FS_RAG—football specific reactive agility test, FS_CODS—football specific change of direction speed test, < 0.2 trivial ES, 0.21–0.49 small ES, 0.50–0.79 medium ES, >0.79 strong ES.

**Figure 3 ijerph-17-00294-f003:**
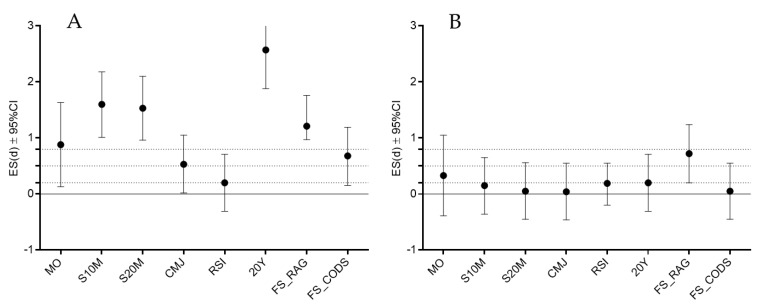
Effect size (ES) differences between performance groups (Starters vs Nonstarters) in U13 (A) and U15 (B) category. LEGEND: MO—maturity offset, S10—sprint 10 meters, S20M—sprint 20 meters, CMJ—countermovement jump, RSI—reactive strength index, 20Y—change of direction speed 20 yards test, FS_RAG—football specific reactive agility test, FS_CODS—football specific change of direction speed test, < 0.2 trivial ES, 0.21–0.49 small ES, 0.50–0.79 medium ES, >0.79 strong ES.

**Table 1 ijerph-17-00294-t001:** Inter-testing reliability of the newly developed tests of reactive agility and change of direction speed (n = 14).

	Test 1		Test 2		Reliability		
Variables	Mean	SD	Mean	SD	ICC	CV	SEM
FS_RAG (s)	2.80	0.21	2.75	0.30	0.79	0.05	0.24
FS_CODS (s)	2.68	0.23	2.65	0.27	0.82	0.04	0.16

LEGEND: FS_RAG—football specific reactive agility test, FS_CODS—football specific change of direction speed test.

**Table 2 ijerph-17-00294-t002:** Intra-testing reliability of the applied tests of fitness capacities in each age category with differences between age categories derived by independent samples t-test.

	U13 (n = 29)				U15 (n = 30)				t-test	
Variables	Mean	SD	Reliability		Mean	SD	Reliability		U13 vs. U15	
			ICC	CV			ICC	CV	t-value	*p*
S10M (s)	2.00	0.14	0.81	0.07	1.88	0.14	0.83	0.06	3.57	0.001
S20M (s)	3.57	0.23	0.84	0.07	3.39	0.18	0.80	0.05	3.13	0.001
CMJ (cm)	24.02	3.54	0.79	0.06	25.80	5.55	0.81	0.06	−1.45	0.15
RSI (index)	99.40	19.33	0.76	0.08	110.70	27.43	0.80	0.07	−1.83	0.07
20Y (s)	5.41	0.20	0.80	0.06	5.10	0.28	0.79	0.06	4.89	0.001
FS_RAG (s)	3.09	0.30	0.78	0.08	2.83	0.17	0.82	0.06	3.96	0.001
FS_CODS (s)	2.93	0.30	0.80	0.05	2.51	0.20	0.85	0.05	6.42	0.001

LEGEND: S10—sprint 10 meters, S20M—sprint 20 meters, CMJ—countermovement jump, RSI—reactive strength index, 20Y—change of direction speed 20 yards test, FS_RAG—football specific reactive agility test, FS_CODS—football specific change of direction speed test, U13—players old 11 and 12 years, U15—players old 13 and 14 years.

**Table 3 ijerph-17-00294-t003:** Descriptive statistics and differences between performance groups in each age category derived by independent sample t-test.

	U13 (n = 29)		U15 (n = 30)	
Variables	Starters	Nonstarters	Starters	Nonstarters
	Mean ± SD	Mean ± SD	Mean ± SD	Mean ± SD
Maturity offset (years)	−0.43 ± 0.22	−0.71 ± 0.39 *	1.33 ± 0.44	1.35 ± 0.59
S10M (s)	1.83 ± 0.15	2.07 ± 0.15 **	1.86 ± 0.13	1.88 ± 0.14
S20M (s)	3.32 ± 0.24	3.68 ± 0.23 **	3.39 ± 0.19	3.40 ± 0.19
CMJ (cm)	26.43 ± 3.65	24.5 ± 3.55 *	26.06 ± 5.55	25.80 ± 6.11
RSI (index)	105.37 ± 19.91	101.4 ± 19.33	109.59 ± 27.34	110.7 ± 24.61
20Y (s)	5.09 ± 0.21	5.63 ± 0.21 **	5.05 ± 0.28	5.10 ± 0.22
FS_RAG (s)	2.81 ± 0.32	3.19 ± 0.31 **	2.51 ± 0.25	2.71 ± 0.30 *
FS_CODS (s)	2.73 ± 0.31	3.05 ± 0.59 *	2.50 ± 0.20	2.51 ± 0.18

LEGEND: S10—sprint 10 meters, S20M—sprint 20 meters, CMJ—countermovement jump, RSI—reactive strength index, 20Y—change of direction speed 20 yards test, FS_RAG—football specific reactive agility test, FS_CODS—football specific change of direction speed test, U13—players old 11 and 12 years, U15—players old 13 and 14 years, * *p* < 0.05, ** *p* < 0.01.

**Table 4 ijerph-17-00294-t004:** Pearson’s product moment correlation coefficients between studied variables in each age category.

		S10M	S20M	20Y	CMJ	RSI	FS_CODS	FS_RAG
S10M	U13	-						
	U15	-						
S20M	U13	0.97 ***	-					
	U15	0.71 *	-					
20Y	U13	0.26	0.30	-				
	U15	0.48 **	0.73 ***	-				
CMJ	U13	−0.59 ***	−0.61 ***	−0.19	-			
	U15	−0.54 **	−0.81 ***	−0.80 ***	-			
RSI	U13	−0.53 **	−0.53 **	−0.27	0.33	-		
	U15	−0.27	−0.43 *	−0.43 *	0.43 *	-		
FS_CODS	U13	0.20	0.23	0.35	0.04	−0.11	-	
	U15	0.09	0.39 *	0.59 ***	−0.59 ***	−0.17	-	
FS_RAG	U13	−0.01	0.07	0.22	0.21	−0.27	0.42 *	-
	U15	0.36	0.49 **	0.64 ***	−0.57 **	−0.33	0.58 ***	-

LEGEND: S10—sprint 10 meters, S20M—sprint 20 meters, CMJ—countermovement jump, RSI—reactive strength index, 20Y—change of direction speed 20 yards test, FS_RAG—football specific reactive agility test, FS_CODS—football specific change of direction speed test, U13—players old 11 and 12 years, U15—players old 13 and 14 years, * *p* < 0.05, ** *p* < 0.01, *** *p* < 0.001.

**Table 5 ijerph-17-00294-t005:** Multiple regression calculation for the football specific change of direction speed, and football specific reactive agility (Criteria) among U13 players.

Criteria	FS_CODS						FS_RAG					
	β	SE (β)	b	SE (b)	t	*p*	β	SE (β)	b	SE (b)	t	*p*
Intercept			−1.88	2.18	−0.86	0.40			0.67	2.17	0.31	0.76
S20M	0.34	0.26	0.44	0.34	1.30	0.20	0.12	0.26	0.16	0.34	0.46	0.65
20Y	0.32	0.19	0.47	0.29	1.65	0.11	0.18	0.19	0.27	0.28	0.96	0.35
CMJ	0.29	0.23	0.02	0.02	1.25	0.22	0.42	0.23	0.04	0.02	1.83	0.08
RSI	0.06	0.22	0.00	0.00	0.28	0.79	−0.29	0.22	0.00	0.00	−1.36	0.19
R	0.44						0.46					
Rsq	0.19						0.21					
*p*	0.25						0.19					

LEGEND: S20M—sprint 20 meters, CMJ—countermovement jump, RSI—reactive strength index, 20Y—change of direction speed 20 yards test, FS_RAG—football specific reactive agility test, FS_CODS—football specific change of direction speed test, Intercept—interception coefficient, R—multiple correlation coefficient, Rsq—coefficient of determination, *p*—level of significance, β—standardized regression coefficient, b—non-standardized regression coefficient, SE—standard error, t—t test value, Note that S10M was not included in multiple regression calculation due to multicollinearity issue.

**Table 6 ijerph-17-00294-t006:** Multiple regression calculation for the football specific change of direction speed, and football specific reactive agility (Criteria) among U15 players.

Criteria	FS_CODS						FS_RAG					
	β	SE (β)	b	SE (b)	t	*p*	β	SE (β)	b	SE (b)	t	*p*
Intercept			2.59	1.45	1.79	0.09			1.75	1.24	1.41	0.17
S20M	−0.35	0.27	−0.38	0.28	−1.32	0.20	−0.09	0.27	−0.08	0.24	−0.31	0.76
20Y	0.45	0.26	0.32	0.18	1.75	0.09	0.51	0.26	0.30	0.16	1.94	0.06
CMJ	−0.56	0.30	−0.02	0.01	−1.89	0.07	−0.21	0.30	−0.01	0.01	−0.70	0.49
RSI	0.11	0.17	0.00	0.00	0.64	0.53	−0.06	0.17	0.00	0.00	−0.36	0.73
R	0.66						0.65					
Rsq	0.35						0.33					
*p*	0.01						0.01					

LEGEND: S20M—sprint 20 meters, CMJ—countermovement jump, RSI—reactive strength index, 20Y—change of direction speed 20 yards test, FS_RAG—football specific reactive agility test, FS_CODS—football specific change of direction speed test, Intercept—interception coefficient, R—multiple correlation coefficient, Rsq—coefficient of determination, *p*—level of significance, β—standardized regression coefficient, b—non-standardized regression coefficient, SE—standard error, t—t test value, Note that S10M was not included in multiple regression calculation due to multicollinearity issue.

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
