# Peer review of "Agility Testing in Youth Football (Soccer)Players; Evaluating Reliability, Validity, and Correlates of Newly Developed Testing Protocols"

_ijerph, 2020, doi:10.3390/ijerph17010294_

Round 1
Reviewer 1 Report
Title: Agility testing in youth football players; evaluating reliability, validity, and correlates of newly developed testing protocols
The experimental design and statistical methods of this study have a certain level, but the writing of the article was not concise. It is recommended to give appropriate streamlining for easy reading.
Similar research on adolescents in the research background and discussion of this article was relatively inadequate. It is recommended to add relative supplements of adolescents research to increase the contribution of this study.
Subject ’s gender, I think it should be male (boy). Please add it in the text.
Please add the subject's physical information, such as height, weight, BMI or percentage body fat. The author has listed BF% in author's another similar work [18], and it is recommended to add relevant information in this article.
The subject's qualifications in football, for example, the time and intensity of training each week, also need to be addressed.
When using regression analysis, what methods were performed (forward, backward, stepwise selection, and related parameters) must be explained.
Compared with adults, the study of adolescents' exercise physiology should consider the subject's biological maturation. As stated by the author in the manuscript " In brief, it is well documented that maturity status in puberty is one of the most important determinants of physical, but also psycho-social development [37].", but the subjects in this article did not have relevant data to illustrate this point. Please add it in the text.
In a similar study of the author's own, football players were analyzed based on different professional positions. Football player roles are different in different positions. There was no description of football players in different positions in this study, which should also be supplemented in this article.
Author Response
Title: Agility testing in youth football players; evaluating reliability, validity, and correlates of newly developed testing protocols
The experimental design and statistical methods of this study have a certain level, but the writing of the article was not concise. It is recommended to give appropriate streamlining for easy reading.
RESPONSE: Thank you for recognizing the quality and potential in our work. Also, thank you for your elaborated comments and suggestions. We tried to follow it specifically and amended the manuscript accordingly. Please see bellow how we responded to your comment and where to find specific amendments. Staying at your disposal.
Similar research on adolescents in the research background and discussion of this article was relatively inadequate. It is recommended to add relative supplements of adolescents research to increase the contribution of this study.
RESPONSE: Indeed, it the original version of the paper we did not pay particular attention to problem of agility in youth, mostly because of the lack of studies which examined the problem of reactive agility in youth athletes. However, following your suggestion in this version of the Introduction, the new paragraph is dedicated specifically to problem of agility in youth and reliability of agility (e.g. CODS) tests in youth age. Text reads: “Because of the known importance of agility for long-term sport development, studies frequently elaborated the reliability and validity of the tests aimed at evaluation of agility in young athletes [17,18]. However, most of the investigations done so far evaluated CODS-, and not RAG- performances in youth athletes involved in different sports, including football. For example, Basque authors examined the reliability and validity of the modified Barrow CODS test and reported high test-retest reliability (ICC: 0.94) for 11-to-14 years old football players [19]. Similarly, Tunisian authors examined zig-zag CODS test was found to be reliable measuring tool for 14.2±0.6 years-old players [20]. Finally, T-drill CODS test had high reliability in 10-12 years old football players [21]. However, there is an evident lack of information about reliability of the RAG tests in football.” (please see 4th paragraph of the Introduction; thank you!).
Subject ’s gender, I think it should be male (boy). Please add it in the text.
RESPONSE: Yes, indeed, we missed this information. Thank you, it is added now (please see 1st sentence of the Participants subsection)
Please add the subject's physical information, such as height, weight, BMI or percentage body fat. The author has listed BF% in author's another similar work [18], and it is recommended to add relevant information in this article.
RESPONSE: More information on participants characteristics are added now. Details of anthropometrics are added. Text reads: “Participants in this study were young football players (all males, n = 59, age: 13.40±1.25 years, body height: 162±8.14 cm, mass: 54.1±9.15 kg, body mass index: 20.57±2.11 kg/m2” (please see Methods, thank you).
The subject's qualifications in football, for example, the time and intensity of training each week, also need to be addressed.
RESPONSE: Thank you for your suggestion. Text is amended and now reads: All participants had at least 3 years of experience in systematic football training, and at the moment of testing participated in 3-4 training sessions weekly, plus one game. Most of their trainings (more than 80%) was oriented toward specific technical and tactical skills, but two of 10 training sessions are oriented toward development of physical and conditioning capacities.” (please see first paragraph of the Participants subsection. Thank you!)
When using regression analysis, what methods were performed (forward, backward, stepwise selection, and related parameters) must be explained.
RESPONSE: The details about multiple regression calculation are added. Text reads: “The standard multiple regressions were calculated to identify the multivariate relationships between independent variables (predictors) and dependent variables (criteria; FS_RAG and FS_CODS). For the purpose of the interpretation of explained variance, the coefficient of multiple correlation, coefficient of determination (percentage of explained variance) were calculated. In order to identify partial contribution of each predictor to overall multiple regression calculation, the standardized- and non-standardized-regression-coefficients were reported. Before multiple regressions’ calculation, the predictors were checked for multicollinearity by calculating the variance inflation factor (VIF), and those variables with a VIF larger than 10 were not included as predictors in the specific multiple regression calculation [33-36].” (Please see 4th paragraph of the Statistics subsection)
Compared with adults, the study of adolescents' exercise physiology should consider the subject's biological maturation. As stated by the author in the manuscript " In brief, it is well documented that maturity status in puberty is one of the most important determinants of physical, but also psycho-social development [37].", but the subjects in this article did not have relevant data to illustrate this point. Please add it in the text.
RESPONSE: Thank you for this comment. In the revised version of the manuscript we included data on maturity offset for the studied sample, based on equations proposed by Moore et al. Text reads For all participants, pubertal timing was estimated according to the biological age of maturity (maturity offset), as described by Moore et al. (Maturity offset = − 7.999994 + (0.0036124 × age (yrs.) × height (cm) [24].” (please see end of 1st paragraph of the Participants subsection). Also, maturity offset is compared across starters and nonstarters in each age group and results are presented in Tables (see Table 2), and Figures (please see Figure 3).
In a similar study of the author's own, football players were analyzed based on different professional positions. Football player roles are different in different positions. There was no description of football players in different positions in this study, which should also be supplemented in this article.
RESPONSE: Thank you for this suggestion. Being honest we intended to apply the position-specific approach in this study as well, but coaches from two football academies included in the investigation convinced us that this is not appropriate for this age. Namely, studied U13 players were not strictly oriented toward specific playing position, and therefore this approach will not be applicable in statistical analyses. However, this is now clearly indicated in the Limitations subsection. Text reads: “Finally, players were not observed according to playing position, which will almost certainly contribute to better applicability of the results. However, U13 players were not strictly defined with regard to their playing-position in football game, which consequently did not allow us to apply such grouping into experimental design.” (please see Limitations and strengths subsection; thank you).
Staying at your disposal
Authors
Reviewer 2 Report
General Comments
The article “Agility testing in youth football players; evaluating reliability, validity, and correlates of newly developed testing protocols” aims to present data regarding specific soccer tests for measuring changing of direction and agility skills of young (U-15 and younger). The article provides innovative and interesting data regarding this topic. In general, it is well-written. The procedures are clearly defined and supported by the literature. There are some minor issues suggested below.
Specific comments
Introduction
The introduction is consistent and provides the most relevant information about the topic of the article. I congratulate the authors for the clear and complete definition and differentiation between agility and change of speed, concepts that are sometimes confused by other authors and readers.
Considering that young players' speed performance (younger than 15 years, like the sample of the current study) is highly constrained by the maturational process, don’t you think that bias is observed when comparing data of players from different ages? In other words, younger players have an incomplete maturational status, so their poor performance in tests is not only explained by their lack of agility or changing of direction ability but determined by their maturation.
Line 90: which specific factors? They were not presented in the introduction, so there is no rationale that supports this analysis.
Methods
It is not clear how the repeated measures were obtained. On the same day? On consecutive days? Please, provide all the details regarding the data collection.
The division of the players between starters and non-starters has no rationale in the introduction. Why did you separate the players? How did you split the players?
Another question regarding this topic is the sample size. When splitting the players into two more groups, you have only 15 (on average) players on each group, so the sample is too small to generalize (but if you have statistically estimated the sample size, you can present this data to support this small sample).
Finally, on this topic, it is not clear the criteria for splitting players. In a normal distribution, the best player of the starter group is probably near the worst player of the non-starter group than the average of the non-starter group (in other words, it is likely to be part of the “population” called non-starter). So, statistically, I disagree with this division.
Line 200: Please, indicate which ICC was used (Consult J Strength Cond Res. 2005 Feb;19(1):231-40. Quantifying test-retest reliability using the intraclass correlation coefficient and the SEM.)
I recommend you present the SEM instead of the CV considering that you calculated the ICC (again, see the abovementioned article).
Results
Nothing to address
Discussion
Line 320: considering the data presented in table 4, I consider that the second hypothesis was denied (most correlations were weak or non-significant).
Interestingly, there are significant and positive associations between FS_RAG and FS_CODS only for the oldest players. This may be explained by the maturational bias presented in younger players. What do you think about it?
Conclusion
There is some discussion included in this topic. I suggest the authors to only conclude the article here, reducing the length of this topic.
Author Response
General Comments
The article “Agility testing in youth football players; evaluating reliability, validity, and correlates of newly developed testing protocols” aims to present data regarding specific soccer tests for measuring changing of direction and agility skills of young (U-15 and younger). The article provides innovative and interesting data regarding this topic. In general, it is well-written. The procedures are clearly defined and supported by the literature. There are some minor issues suggested below.
RESPONSE: Thank you for recognizing the potential and importance of our work. Also, we are particularly grateful to your comments and suggestions. Please see how we responded, and where to find specific changes. Staying at your disposal.
Specific comments
Introduction
The introduction is consistent and provides the most relevant information about the topic of the article. I congratulate the authors for the clear and complete definition and differentiation between agility and change of speed, concepts that are sometimes confused by other authors and readers.
RESPONSE: Thank you for recognizing our efforts in explaining the problem.
Considering that young players' speed performance (younger than 15 years, like the sample of the current study) is highly constrained by the maturational process, don’t you think that bias is observed when comparing data of players from different ages? In other words, younger players have an incomplete maturational status, so their poor performance in tests is not only explained by their lack of agility or changing of direction ability but determined by their maturation.
RESPONSE: Off course, we agree. It seems that we did not pay attention that this problem should be emphasized in more details. We tried to correct it now. Specifically, at certain places in manuscript the problem of maturity, and influence of maturity on physical fitness capacities is highlighted. For example, “While maturation level is an important determinant of overall conditioning status in the younger age group (U13), etc.” (last paragraph of the subsection 4.2); “Therefore, it can be concluded that, while physical and maturity-related variables are more important determinants of success in younger age, with the advancement of the maturity process, the importance of skill-related variables increases.” (2nd paragraph of the Conclusion), etc.
Line 90: which specific factors? They were not presented in the introduction, so there is no rationale that supports this analysis.
RESPONSE: Thank you for noticing it. We thought “conditioning capacities”, and the problem of its influence on RAG and CODS is now accentuated in the 2nd paragraph of the Introduction (e.g. “Indeed, while various conditioning capacities (i.e. sprinting speed, jumping performance) were more strongly associated with CODS than with RAG, perceptual and cognitive capacities were identified as being significant predictors of RAG in team sport athletes [14,15]. Meanwhile, there is an evident lack of studies which examined factors associated with CODS and RAG in football. These information may be of high importance in training and conditioning since it will allow specific and targeted development of important qualities which will consequently improve specific RAG or CODS quality [16].”) Thank you.
Methods
It is not clear how the repeated measures were obtained. On the same day? On consecutive days? Please, provide all the details regarding the data collection.
RESPONSE: Thank you for the comment. It is specified now, and text reads: “The repeated measurement (test and retest were separated by 7-9 days) was applied in order, etc.” (please see 2nd paragraph of the Study design, thank you!)
The division of the players between starters and non-starters has no rationale in the introduction. Why did you separate the players? How did you split the players?
RESPONSE: The grouping into starters and non-starters is now explained. Text reads: “The grouping into starters and non-starters was done by coaches (based on their experience during competitions), and investigators had no influence on coaches’ decision.” (Please see highlighted text in 1st paragraph of the Participants subsection).
Another question regarding this topic is the sample size. When splitting the players into two more groups, you have only 15 (on average) players on each group, so the sample is too small to generalize (but if you have statistically estimated the sample size, you can present this data to support this small sample).
RESPONSE: Indeed, when the sample was divided in two subsgroups (starter vs nonstarters) the sample is small. However, because of that reason, we calculated effect size differences as well and did not rely solely of differences calculated by t-test. Also, according to 3rd Reviewer’s suggestion, in this version of the manuscript we included graphical presentations of normality of distributions for observed variables (please see Supplementary Figure). Thank you.
Finally, on this topic, it is not clear the criteria for splitting players. In a normal distribution, the best player of the starter group is probably near the worst player of the non-starter group than the average of the non-starter group (in other words, it is likely to be part of the “population” called non-starter). So, statistically, I disagree with this division.
RESPONSE: Actually, the situation is opposite. The “best player” in non-starter group is near the “worse player” in starter group. It seems that under-explained the fact that “starters” are “those who start the game (best players according to coaches’ opinion). For that reason we specified it more precisely in Methods. Text reads: "The grouping into starters and non-starters was done by coaches (based on their experience during competitions), and investigators had no influence on coaches’ decision. Specifically, starters were players who regularly “start the game” (i.e. better players as defined by their coaches), and non-starters were actually their “substitutes”. (please see Participants subsection)
Line 200: Please, indicate which ICC was used (Consult J Strength Cond Res. 2005 Feb;19(1):231-40. Quantifying test-retest reliability using the intraclass correlation coefficient and the SEM.)
RESPONSE: Thank you, more details are added. Also, SEM is calculated. Text reads: “The inter-testing reliability of the FS_CODS and FS_RAG was analyzed by the intraclass correlation coefficient (ICC; model 3,1) as a measure or relative reliability, and by the coefficient of variation (CV) as a measure of absolute reliability [28-30]. Note that we included the specified reference (Weir 2005). Thank you!
I recommend you present the SEM instead of the CV considering that you calculated the ICC (again, see the abovementioned article).
RESPONSE: As you suggested, in this version of the manuscript the standard error of measurement (SEM) was also calculated (SEM= SD- √(1-ICC)).” (please see 2nd paragraph of the Statistics). Also, SEM values are reported in table 1. Thank you
Results
Nothing to address
RESPONSE: Thank you.
Discussion
Line 320: considering the data presented in table 4, I consider that the second hypothesis was denied (most correlations were weak or non-significant).
RESPONSE: Thank you, we accepted your suggestion and denied the 2nd hypothesis of the study. Text reads: “Collectively, we may accept our first study hypothesis (that FS_CODS and FS_RAG have appropriate reliability and validity), while because of the week correlations between studied conditioning capacities with FS_CODS and FS_RAG, the second study hypothesis is denied. “ (please see 1st paragraph of the Discussion).
Interestingly, there are significant and positive associations between FS_RAG and FS_CODS only for the oldest players. This may be explained by the maturational bias presented in younger players. What do you think about it?
RESPONSE: Indeed, the correlation between CODS and RAG is stronger in U15 (but also significant in U13), and this is almost certainly a results of maturational bias. This is briefly discussed now. Thank you. Text reads: “The previous discussion may be additional supported by the fact that correlation between FS_CODS and FS_RAG is stronger in U15 than in U13 (Pearson’s r =0.42 and 0.58, respectively), and this is certainly related to maturation process” (please see end of 3rd paragraph of the Validity subsection).
Conclusion
There is some discussion included in this topic. I suggest the authors to only conclude the article here, reducing the length of this topic.
RESPONSE: We tried to shorten the Conclusions section, but we weren’t able to shorten it to a greater extent since 3rd Reviewer asked for additional details about practical applicability of results (please see last two paragraphs). However, if you will insist on further reduction, we will certainly do our best to meet your demands. Thank you.
Staying at your disposal
Authors
Reviewer 3 Report
The work concerns the important cognitive and application problem of agility testing in young footballers. I rate the article positively, but I have a few critical remarks. In the methodological part of the article too little space is devoted to the description of the research procedure. This applies particularly to the method of selecting the sample for the study and the date (and time of day) in which the study was conducted. The description of the research material should also be supplemented with information on average age values ​​and somatic features, separately for each of the examined groups and the possible significance of differences between them. There is also no information on the distribution of the analyzed variables. The nature of the research sample and the methods of statistical analysis used lead to the conclusion that the distribution of variables was not consistent with the normal distribution. Consequently, the use of arithmetic means and standard deviations in the article should be considered incorrect. Medians and quarter deviations had to be used. The results of the regression analysis were not statistically significant, therefore I also recommend softening the final conclusions. The authors pointed to applicability as a strength of work. However, there is no information about what new work brings to science and what research gap it closes.
Author Response
Reviewer 3
The work concerns the important cognitive and application problem of agility testing in young footballers. I rate the article positively, but I have a few critical remarks. In the methodological part of the article too little space is devoted to the description of the research procedure. This applies particularly to the method of selecting the sample for the study and the date (and time of day) in which the study was conducted.
RESPONSE: Thank you for your positive rating of the article. We tried to follow your suggestions and improved the certain parts of the article you specified. Collectively, your suggestion are pretty similar to those of other 2 reviewers so we gladly accepted it all. With regard to your previous comment (participants) details are added. Text (with regard of testing details) now reads: “The testing was done during March 2019, and all to avoid diurnal variations, all tests were performed in the morning hours between 10:00 and 12:00.” (please see last sentence of Participants subsection). Also, details for inclusion of the observed football academies are included, and text reads: “Football academies included in this study were selected on the basis of following criteria: (i) being involved in football training for age categories starting from U11 team to seniors (+18 years), (ii) having at disposal courts with artificial-plastic turf (see later for details about testing), and (iii) licensed (professional) coaches working with U13 and U15 teams (see later for details).” (please see 1st paragraph of the Participants subsection).
The description of the research material should also be supplemented with information on average age values ​​and somatic features, separately for each of the examined groups and the possible significance of differences between them.
RESPONSE: In this version of the manuscript we included data on maturity offset (i.e. peak height velocity) are included. Also, differences between starters and non-starters are evaluated in this parameter, and results are presented in Table 3, and Figure 3. Results are consequently used in Discussion section. For example, text in subsection 4.2. “Therefore, it is not surprising that starters in the younger age group (12–13 years of age) were biologically advanced in maturation status (please see differenced in maturity offset), and outperformed the non-starters in all studied capacities, including FS_RAG and FS_CODS. The previous discussion may be additional supported by the fact that correlation between FS_CODS and FS_RAG is stronger in U15 than in U13 (Pearson’s r =0.42 and 0.58, respectively), and this is certainly related to maturation process. While maturation level is an important determinant of overall conditioning status in the younger age group (U13), etc.”
There is also no information on the distribution of the analyzed variables. The nature of the research sample and the methods of statistical analysis used lead to the conclusion that the distribution of variables was not consistent with the normal distribution. Consequently, the use of arithmetic means and standard deviations in the article should be considered incorrect. Medians and quarter deviations had to be used.
RESPONSE: Thank you for this suggestion. The distributions were checked for normality by Kolmogorov Smirnov test, and results now presented in Supplementary figure 1. However, if you will insist on reporting Medians and QDev we will certainly meet your requirements. Staying at your disposal.
The results of the regression analysis were not statistically significant, therefore I also recommend softening the final conclusions.
RESPONSE: Actually, the regression analysis was significant for U15 (please see Table 6). However, since you suggested that Discussion should be “softened”, we followed your suggestion and amended the Conclusion accordingly.
The authors pointed to applicability as a strength of work. However, there is no information about what new work brings to science and what research gap it closes.
RESPONSE: Thank you for noticing it. We tried to “strengthen” the report on applicability of obtained and discussed results. Text reads: ” From the practical aspect it is important to note that that our results point that improvement in sprint and jumping could be theoretically beneficial in improvement of sport-specific RAG and CODS in U15 players. Meanwhile, the RAG and CODS in U13 players are more influenced by various neurological factors, including technique and quality of the execution of the specific movement templates that occur in the applied testing sequence. As a result, training programs aimed at the improvement of RAG and CODS in early pubescent football players should be more oriented toward achieving an accurate and effective movement technique, and not toward the development of conditioning capacities, which (theoretically) contribute to RAG and CODS due to physiological background (i.e., the necessity of the fast development of force).” (Please see last paragraph of the Conclusion section).
Staying at your disposal
Authors
Round 2
Reviewer 1 Report
Thanks for the author ’s intention to reply and correct the suggestion. The study should have met the basic requirements for publication in the scientific literature. My personal suggestion can be published on IJERPH.